# Enhancing Privacy and Utility in Differentially Private Deep Learning Through Instance-Level Smoothing

## Abstract

In this paper, we address the dual challenge of maintaining high accuracy and ensuring fairness in differentially private (DP) deep learning models. The optimization process is inherently complicated by the necessity of injecting random noise and limiting training iterations, particularly for over-parameterized models. Moreover, DP mechanisms frequently exacerbate accuracy disparities across subpopulations, complicating the balance between privacy and fairness. To tackle these challenges, we introduce a novel framework that systematically addresses the trade-off between privacy and utility in DP deep learning. At the core of our approach is the concept of instance-level smoothing, which enhances privacy protections without compromising performance. Our theoretical contributions include deep insights into sample complexity, instance-level smoothing factors, and error bounds required to achieve a given privacy budget. These insights provide a robust foundation for optimizing the delicate balance between privacy and utility. Our method demonstrates remarkable robustness, independent of iteration counts, model parameters, batch normalization processes, and subpopulation disparities. This flexibility enables an optimal balance between privacy preservation and utility, adaptable to a wide range of scenarios. Through extensive empirical studies on the large-scale medical imaging dataset CheXpert, we validate the effectiveness of our approach. Our findings align with theoretical predictions, showing that our method can effectively meet stringent privacy requirements while maintaining high performance. By bridging the gap between formal privacy guarantees and practical deep learning applications, our work lays the groundwork for future advancements in the field. This research empowers practitioners to protect sensitive data during model training and ensures both data privacy and model generality, paving the way for more secure and equitable AI systems.

## 1 Introduction

Large neural networks with billions of parameters have achieved state-of-the-art performance across a wide range of machine learning tasks Brown & Mann (2020); Zhai et al. (2022). Despite their success, these models are known to memorize training data Zhang et al. (2021), leading to potential leakage of sensitive information. Privacy attacks capable of extracting memorized training data have been demonstrated on various types of models, including language models Carlini et al. (2021), diffusion models Carlini et al. (2023), and image classification models Balle et al. (2022). Additionally, membership inference attacks, which can determine whether a particular data point was used to train a model, have proven successful across multiple architectures and data modalities Carlini et al. (2022). The implications of these vulnerabilities are profound, particularly in applications involving private, confidential, or proprietary data, such as healthcare, finance, recommendation systems, and mobility. To ensure the safe deployment of models trained on such data, it is crucial to address these privacy concerns effectively.

Differential Privacy (DP) Dwork et al. (2006) has emerged as the gold standard for protecting individual privacy in data processing algorithms. However, achieving strong privacy protections with DP is particularly challenging in deep learning, especially as model sizes and data dimensions increase. The most popular DP training technique in deep learning is Differentially Private Stochastic Gradi-

ent Descent (DP-SGD) Abadi et al. (2016), which privatizes gradients by clipping and adding noise. The strength of the privacy guarantee $\epsilon$ depends on the noise scale, batch size, number of training samples, and number of iterations. While DP-SGD is a near drop-in replacement for standard SGD, it presents significant challenges that have hindered its widespread adoption.

Achieving high accuracy with differentially private (DP) models faces two major challenges. First, injecting random noise and limiting training iterations complicates optimization, as noisy gradients and iteration constraints hinder effective hyperparameter tuning Anil et al. (2021). The noise's Euclidean norm scales with model size, degrading performance for larger models Yu et al. (2021). This issue is particularly problematic given the prevalence of over-parameterized neural networks in AI. Second, fairness issues arise in deep neural networks, exacerbated by DP mechanisms. Models can show accuracy disparities across subpopulations, such as race, age, sex, or insurance type in medical tasks Seyyed-Kalantari et al. (2021). DP can worsen these disparities, increasing accuracy gaps due to unbalanced subgroup data Bagdasaryan et al. (2019); Santos-Lozada et al. (2020). DP limits information extraction from individual data points, conflicting with the accurate learning needed for small or underrepresented subgroups, particularly in medical datasets Cummings et al. (2019); Suriyakumar et al. (2021).

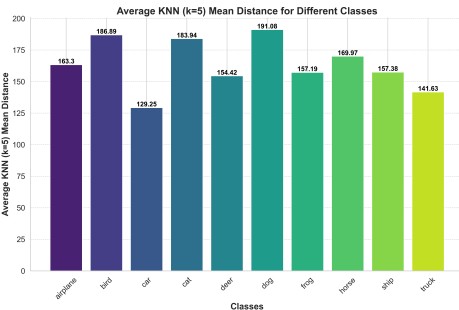

Figure 1: Histogram of the mean distances for k-nearest neighbors (KNN) in CIFAR10. This histogram visualizes the distribution of average distances among data points as computed by the KNN algorithm.

Understanding the trade-off between privacy and utility in differentially private (DP) deep learning models is crucial for both theoretical and practical advancements. This paper introduces a novel framework that aims to bridge the performance gap between DP models and their non-private counterparts by training an end-to-end DP network. Our central hypothesis is that samples with sparse distributions in their feature space, based on pre-trained large models, are more prone to privacy leakage. In contrast, classes with sufficient samples that are compactly clustered in the feature space will be well-represented, incurring minimal privacy sacrifice while maintaining high performance.

To illustrate the difference in feature density among classes in the CIFAR-10 dataset, we present a graph depicting the mean distances within the 10 classes. This graph measures the proximity of each sample to its 5 nearest neighbors. As shown in Figure 1, the mean sample distance for *car* is 129.25, whereas for *dog*, it is 191.08. The observed trade-off between privacy and utility Berrada et al. (August 2023) aligns well with these mean distances: the *car* class exhibits the best classification accuracy and the least privacy leakage, whereas the *dog* class suffers from significant privacy leakage and class disparity issues.

Given that each class has an equal number of 5000 training samples, the within-class compactness and between-class distance largely determine the privacy-utility trade-off. In scenarios where the sample numbers across classes are imbalanced, privacy preservation and class disparity issues worsen. This observation motivates us to consider instance-level probability density function (PDF) estimation and instance-specific kernel smoothing to enhance privacy without degrading performance. We prefer the term *instance smoothing* over *noise*, as it better captures the essence of our approach in providing privacy protection alongside model generality.

Our method remains flexible and can incorporate future advancements in deep learning, including improved network architectures and pre-trained models, without additional constraints. This work

represents a significant step towards enabling deep learning practitioners to leverage formal privacy guarantees offered by DP while protecting sensitive data during model training.

The contributions of this paper are multifaceted and significant, paving the way for a new paradigm in differentially private deep learning:

- **Innovative Framework:** We introduce a novel framework that systematically addresses the trade-off between privacy and utility in differentially private (DP) deep learning models. Central to our approach is the concept of instance-level smoothing and PDF representation learning, which significantly enhances privacy without compromising model performance.

- **Theoretical Insights:** Our work provides profound theoretical insights into key aspects such as sample complexity, instance-level smoothing factors, and error bounds for achieving a given privacy budget. These insights offer a robust foundation for understanding and optimizing the balance between privacy and utility.

- **Robust Methodology:** The proposed method demonstrates remarkable robustness, being independent of iteration counts, backbone parameters, batch normalization processes, and sub-class disparities. This flexibility allows for an optimal balance between privacy preservation and utility, making it adaptable to a wide range of scenarios.

- **Empirical Validation:** Through extensive empirical studies on the large-scale medical image dataset CheXpert Irvin et al. (January 2019), we validate the effectiveness of our approach. Our findings confirm the theoretical predictions, demonstrating that our method can effectively meet privacy requirements while maintaining high performance. This new paradigm has the potential to significantly expand the use of large models in real-world scenarios where privacy is paramount.

## 2  RELATED WORKS

**Zero-Concentrated Differential Privacy**    Differential Privacy (DP) was initially formalized by Dwork et al. (2006) and later adapted to deep learning by Abadi et al. (2016), who operationalized DP-SGD Bassily et al. (2014) to train neural networks with differential privacy guarantees. The central goal in differentially private learning is to train a classifier while satisfying a rigorous mathematical definition of privacy known as differential privacy Dwork (2006), which ensures that any individual training data point cannot be identified using the trained model and any additional side information. More formally, we adopt the popular modern variant called zero-centered Concentrated Differential Privacy (zCDP), as defined below:

**Definition 1.** ZERO-CONCENTRATED DIFFERENTIAL PRIVACY BUN & STEINKE (2016)
*Two datasets $D_0$ and $D_1$ are neighbors if they can be constructed from each other by adding or removing one data point. A randomized mechanism $\mathcal{A}$ satisfies $\rho$-zero-concentrated differential privacy ($\rho$-zCDP) if, for all neighboring datasets $D_0$ and $D_1$, we have*

$$R_\alpha(\mathcal{A}(D_0)\|\mathcal{A}(D_1)) \leq \rho\alpha,$$

*where*

$$R_\alpha(P\|Q) = \frac{1}{\alpha - 1} \log \int \left(\frac{p(x)}{q(x)}\right)^\alpha q(x)\, dx$$

*is the Rényi divergence between two distributions $P$ and $Q$.*

In the above definition, $\rho \geq 0$ is the privacy loss parameter that measures the strength of the protection. $\rho = 0$ indicates perfect privacy, while $\rho = \infty$ means no protection at all. The privacy protection is considered sufficiently strong in practice if $\rho$ is a small constant, e.g., 1, 2, 4, 8. For readers familiar with standard approximate DP but not zCDP, $\rho$-zCDP implies $(\epsilon, \delta)$-DP for all $\delta > 0$ with $\epsilon = \rho + 2\sqrt{\rho \log(1/\delta)}$.

While DP-SGD Berrada et al. (August 2023) is foundational for differential privacy in deep learning, it has significant drawbacks. It requires a trade-off between privacy budget ($\epsilon$), noise addition, and the number of iterations, impacting model utility. The method is sensitive to hyperparameters and regularization, and per-sample gradient clipping introduces bias and high variance, degrading performance. Additionally, the noise scales with model dimension, complicating its application to large models and disrupting batch normalization.

We propose a novel approach that bypasses these issues. Our method adjusts the smoothing bandwidths of the PDF per instance, independent of iterations, enhancing the balance between privacy and accuracy. It is robust to hyperparameter settings and avoids gradient clipping and noise addition, reducing bias and variance. This allows for fine-tuning large pre-trained models in high-dimensional spaces without affecting batch normalization. We also provide theoretical bounds on sample size and classification errors, demonstrating its effectiveness.

Overall, our method offers a practical and efficient alternative to DP-SGD, maintaining privacy while improving model performance.

**Adaptive Kernel Density Fine-tuning** Our work diverges from the DP-SGD method by drawing on adaptive kernel metric representation learning, particularly useful for tasks like anomaly detection. Kernel density-based learning captures the underlying probability density of data, using Gaussian kernels to assess local density and identify anomalies, which are prone to privacy risks in sparse regions Zhang et al. (2018). Kernel-based methods handle uncertainty and class-specific variances well, as demonstrated by Non-isotropic von Mises-Fisher (nivMF) distributions that model complex variances to enhance generalization Kirchhof et al. (2022).

We employ a projection network parameterized by $W \in \mathbb{R}^{p \times d}$, where $d$ is the feature dimension from the backbone network, and $p$ is the projected dimension in the PDF space. This adaptive kernel estimation-based approach emphasizes local data density, crucial for identifying and mitigating privacy risks in sparse regions. It allows fine-tuning of pre-trained models in high-dimensional spaces, compatible with any backbone architecture.

Unlike DP-SGD, which requires per-sample gradient clipping and noise addition, our method adjusts the smoothing bandwidth at the instance level, leading to more efficient training. By focusing on instance-level smoothing, we effectively balance privacy and accuracy, offering a practical and efficient alternative to DP-SGD for differentially private learning.

## 3  METHOD

Our training framework for medical images starts by feeding samples into feature extraction networks, such as pre-trained transformers on ImageNet, to obtain initial feature representations. We then project these raw features into a PDF representation space, modeling each class with an independent PDF using an adaptive kernel density estimation method. This approach assigns a specific smoothing bandwidth to each instance based on its relative distribution density.

As shown in Figure 2, we construct a PDF for each class by applying adaptive kernel density estimation in the projected space, ensuring customized privacy preservation for each instance. To maintain compactness within classes and distinctiveness between different classes, we employ a PDF contrastive loss with a margin. This loss function encourages the model to achieve the desired feature separation. During testing, we compute the likelihood of a test sample belonging to each class by comparing it against the trained PDFs. The predicted label is assigned based on the class with the highest probability. The detailed steps are depicted in Figure 2 and will be elaborated in the following sub-sections.

### 3.1  INSTANCE LEVEL PRIVACY BUDGET ALLOCATION

This projection network can map the samples in feature space in $\mathbb{R}^d$ to PDF in $\mathbb{R}^p$. For example, in our experiment settings, $d = 768$ is for transformer and $p = 64$ for the embedding PDF space. The parameter $p$ is selected by evaluation process.

Each training sample exhibits different local sensitivity in the PDF space. We should allocate the privacy budget according to their sensitivity. Adaptive smoothing mechanism adjusts the amount of bandwidth based on the local sensitivity of the PDF estimate. This approach enhances privacy while maintaining the utility of the PDF representation.

**Proposition 1.** INSTANCE LEVEL PRIVACY SMOOTHING
*Let $x_i \in \mathbb{R}^p$ be a sample. To protect each sample $x_i$, smoothing factor $N_i$ is applied for each dimension, where $N_i \sim \mathcal{N}(0, s_i^2 \cdot \sigma^2)$. Here, $s_i$ represents the local sensitivity of $x_i$, and $\sigma$ is a global privacy setting applicable to all training samples. The protected sample $\tilde{x}_i$ is then given*

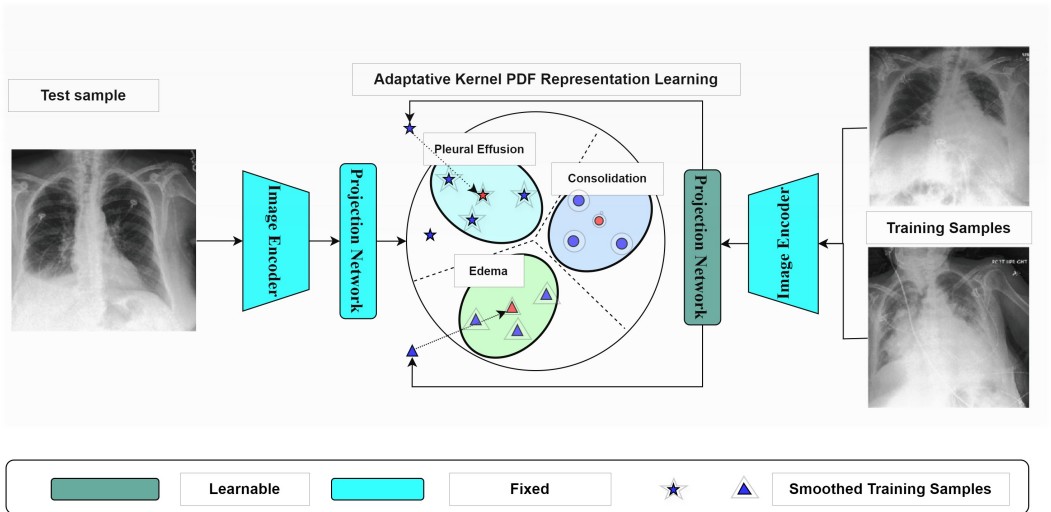

Figure 2: Adaptive Kernel PDF Representation Learning Framework. In the first stage, each training sample is smoothed with a customized bandwidth: densely distributed samples receive less smoothing, while sparsely distributed samples are protected with larger bandwidths. In the second stage, the projection network is trained to optimally separate each class. The mean vectors for each class are represented by red markers: stars, circles, and triangles correspond to the classes *Pleural Effusion*, *Edema*, and *Consolidation*, respectively, positioned at the centers of their corresponding elliptical PDFs. Each class is formed as a compacted PDF with a margin among other classes. The varying sizes of the shaded areas around the samples represent different local bandwidths, corresponding to varied probability densities and different allocations of privacy budgets.

*by $\tilde{x}_i = x_i + N_i$. This mechanism ensures that the level of smoothing corresponds to the local sensitivity of each data point, balancing privacy protection and data utility by scaling the smoothing variance with $s_i^2 \cdot \sigma^2$. The introduction of smoothing $N_i$ adheres to differential privacy principles, providing a customizable level of privacy suited to the sensitivity of the data.*

We will define the $s_i$ firstly. The PDF for the training set is defined as:

$$\hat{p}(x) = \frac{1}{n} \sum_{j=1}^{n} K_h(x - x_j)$$

.

The PDF without sample $x_i$ is:

$$\hat{p}_{-i}(x) = \frac{1}{n-1} \sum_{j \neq i} K_h(x - x_j)$$

Then the local sensitivity $s_i$ at $x_i$ is:

$$s_i = |\hat{p}(x) - \hat{p}_{-i}(x)|$$

For large $n$, this can be approximated as:

$$s_i \approx \frac{1}{n} K_h(x - x_i)$$

The $K_h$ is the kernel function with smoothing bandwidth $h$, such as the Gaussian Kernel. However, in this paper, we will define the Adaptive Kernel Function in the next sub-section to best describe each sample's distinctiveness.

The $\tilde{x}_i = x_i + N_i$ provides instance level privacy protection mechanism. The modified PDF estimate $\hat{p}(x)$ is:

$$\hat{p}(x) = \frac{1}{n} \sum_{i=1}^{n} K_h(x - \tilde{x}_i)$$

In the following section, we will not use $\tilde{x}_i$ but $x_i$ for brevity to represent the enhanced samples. The customized smoothing factor $N_i$ is related to instance sensitivity $s_i$ and an global smoothing variance $\sigma^2$ independent of sample index.

**Theorem 1.** PRIVACY-PRESERVING FOR $\rho$-ZCDP
*To achieve $\rho$-zero Concentrated Differential Privacy ($\rho$-zCDP), the smoothing variance is chosen as $\sigma^2 = \frac{2\ln(1/\delta)}{\rho}$. The privacy-preserving parameter $\rho$ controls the overall level of privacy, ensuring that the samples in PDF space remains stable and robust while preserving privacy. This formulation provides a balance between privacy protection and data utility by adjusting the smoothing variance according to the specified privacy parameter $\rho$.*

The proof can be found in Appendix A.1.

## 3.2 ADAPTIVE KERNEL IN HIGHER DIMENSIONS

**Proposition 2.** ADAPTIVE KERNEL SMOOTHING
*In the context of the higher-dimensional adaptive kernel, we can define $h'$ explicitly based on the local density, which provides additional privacy-preserving ability for the sparse training samples given the base bandwidth $h$. The adaptive kernel $K_h(x)$ in $p$ dimensions is given by:*

$$K_h(x) = \begin{cases} \frac{1}{(2\pi h^2)^{p/2}} \exp\left(-\frac{\|x\|^2}{2h^2}\right) & \text{if } N_r(x) \geq \lambda, \\ \frac{1}{(2\pi(h^2+h'^2))^{p/2}} \exp\left(-\frac{\|x\|^2}{2(h^2+h'^2)}\right) & \text{if } N_r(x) < \lambda, \end{cases}$$

*where $h' = h \cdot \frac{c}{N_r(x)}$. To define sparse and dense regions in high-dimensional space, we use a measure of local density, for which one effective approach is the $k$-nearest neighbors ($k$-NN) method.*

To distinguish between dense and sparse regions, we set a threshold $\lambda$. This threshold can be determined based on the empirical distribution of $k$-NN distances in the dataset. For example, setting $\lambda$ to the median or the 75th percentile of $k$-NN distances ensures a robust boundary. Formally, we define regions as follows:

$$\begin{cases} \text{Dense region} & \text{if } N_r(x) \geq \lambda \\ \text{Sparse region} & \text{if } N_r(x) < \lambda \end{cases}$$

where $\lambda$ is a chosen percentile of the distribution of $k$-NN distances.

For example, in the CIFAR10 dataset, the mean 5-nearest-neighbor distance for *car* is 129.25. We can define $\lambda = 129.25$. For samples with mean distance to their 5-nearest-neighbor beyond this threshold, we should use extra bandwidth to protect the samples.

**Theorem 2.** SAMPLE SIZE COMPLEXITY
*To preserve $\rho$-zCDP in PDF space, the number of samples $n$ should satisfy:*

$$n \geq \frac{C}{\rho(\alpha - 1)}$$

*This theoretical bound ensures that the change in the KDE estimate due to removing a single sample is sufficiently small to maintain the desired privacy level $\rho$. The constant $C$ is given by:*

$$C = \frac{1}{\sqrt{2\pi h^2}}$$

*This bound provides a theoretical guideline for the minimum number of samples needed to ensure $\rho$-zCDP in the context of PDF probability density estimation.*

The proof can be found in Appendix A.2.

## 3.3 KERNEL DENSITY-BASED REPRESENTATION LEARNING

In the training framework, the pre-trained baseline network are frozen, which provides feature representation ability thanks to their extensive pre-training on large public datasets such as ImageNet. We define $\mathbf{W}^\top \in \mathbb{R}^{d \times p}$ serves as the projection matrix that transforms the raw feature space of training samples to the PDF space, facilitating comparisons among different classes.

We propose our learning objective to act as the training loss, replacing the conventional cross-entropy loss and guiding the network training process.

$$
\begin{aligned}
\mathcal{L}(\mathbf{W}) = \max(&-\sum_{x \in D_t} \mathbb{1}_{\{y_b = t\}} \mathbf{K}(x - x_b) \\
&+ \sum_{x \in D_t} \mathbb{1}_{\{y_b \neq t\}} (\mathbf{K}(x - x_b) + m, 0)
\end{aligned}
\tag{1}
$$

In Eq. 1, the set of trainable parameters is denoted by $\mathbf{W}$, which is implemented as a projection network in our design. In this context, $x$ represents the PDF representations of all training sample, and $\mathbf{x}_b$ signifies a batch of instances, while the $y_b$ is the class label for $\mathbf{x}_b$. The symbol $m$ is introduced as a PDF margin, ensuring that the PDF for positive samples with the same $y_b$ exceeds that of negative instances with different class label by a safe margin. Eq. 1 is employed as the contrastive loss function within our framework. Notably, the probability values involved in the computation are expressed in logarithmic format. This approach not only stabilizes the training process but also prevents the values from experiencing underflow during back-propagation.

For the PDF-based classification model, if a test sample is nearer to a class in the PDF representation, the sample should be classified to the given class. Assuming there are $k$ classes, we need to incorporate the privacy-preserving mechanism into the accuracy calculation.

For a sample $x$, compute the KDE estimates for each class $j \in \{1, \ldots, k\}$:

$$
\hat{p}_j(x) = \frac{1}{n_j} \sum_{i=1}^{n_j} K_h(x - x_{ij})
$$

where $n_j$ is the number of samples in class $j$ and $x_{ij}$ are the samples belonging to class $j$. Assign the sample $x$ to the class with the highest KDE estimate:

$$
\hat{y}(x) = \arg\max_j \hat{p}_j(x)
$$

**Theorem 3.** BOUNDS ON MISCLASSIFICATION ERRORS
*The accuracy of KDE-based classification is affected by the amount of smoothing added to achieve $\rho$-zCDP. The trade-off between privacy and accuracy can be analyzed using the following bound for a projection network $f : \mathbb{R}^d \to \mathbb{R}^p$ such that the misclassification error:*

$$
\mathrm{Err}(f) \leq 2 \exp\left(-\frac{m^2 \rho}{4 \ln(1/\delta)}\right)
$$

*Higher $\rho$ leads to more smoothing and thus lower classification accuracy. Balancing $\rho$ appropriately ensures a reasonable trade-off between privacy and accuracy. The $m$ is the PDF margin in the training process.*

The proof can be found in Appendix A.3.

## 4 EMPIRICAL STUDIES

### 4.1 DATASETS

The CIFAR10 dataset Krizhevsky (2009) comprises 32×32 color images across 10 distinct classes, encompassing 50,000 training examples and 10,000 testing examples. We train the model using the proposed privacy-preserving mechanism. This setup is crucial for demonstrating the effectiveness of privacy-preserving techniques in image classification tasks, ensuring that sensitive data is

safeguarded during the training process. We further conduct experiments on the real medical CheX-pert dataset Irvin et al. (January 2019), a substantial collection comprising 224,316 chest X-rays from 65,240 patients. This dataset includes five classes representing various thoracic pathologies: (a) Atelectasis, (b) Cardiomegaly, (c) Consolidation, (d) Edema, and (e) Pleural Effusion. For our purposes, we reinitialize only the projection layer while keeping the other layers fixed, ensuring no impact on privacy leakage as described in Abadi et al. (October 2016).

## 4.2 EXPERIMENT SETTINGS

To evaluate both non-private and private deep learning models, we employ the area under the curve (AUC) metric. For our model, we set the projected PDF dimension to 64, as this configuration offers the best balance between prediction accuracy and training efficiency. The PDF margin in our learning process is set to 10 (in logarithmic scale) based on cross-validation on the CIFAR10 dataset Krizhevsky (2009). For the differentially private stochastic gradient descent (DP-SGD) methods, the training settings are configured according to the guidelines provided in Berrada et al. (August 2023).

An algorithm that satisfies $\rho$-zCDP also satisfies $(\epsilon, \delta)$-DP, where:

$$\epsilon = \rho + 2\sqrt{\rho \log(1/\delta)}$$

Based on this relationship, we will compare our method with other approaches under the same pre-trained models, such as ResNet-50, ViT-B/16, and DenseNet. In our experimental settings, the broken probability $\delta$ is fixed at $10^{-5}$. It is important to note that in this context, $\delta$ denotes the broken probability, which is distinct from the $\delta$ used in the previous section to represent the confidence bound.

We will assess how varying $\epsilon$ values in $(\epsilon, \delta)$-DP affect model performance, corresponding to different $\rho$ values in $\rho$-zCDP. This evaluation will provide insights into the trade-offs between privacy guarantees and model efficacy.

## 4.3 RESULTS AND DISCUSSION ON CIFAR10

To validate the effectiveness of the proposed method and the theoretical analysis presented in Section 3, we conducted experiments using the CIFAR-10 dataset. The DP-SGD methods adhere to the settings outlined in Berrada et al. (August 2023), with the primary difference being the backbone models used: ResNet18, ResNet-50, and ViT-B/16. All models were pre-trained on ImageNet-21K. For the Non-Private method, we utilized the optimal settings without enforcing any smoothing. Several observations can be drawn from the results. The batch size was set to 4096 to favor DP-SGD, although we did not explore extremely large batch sizes due to computational limitations.

**Model Capacity and Smoothing Resilience**   our method exhibits a narrow gap with the Non-Private model, lagging by just 2-3 percent when $\epsilon = 8$. However, for the DP-SGD method, even under the setting of $\epsilon = 8$, its performance is still inferior to our $\epsilon = 3$ counterpart. This indicates that our method provides significantly better utility for a given privacy budget. Our experiments reveal that larger models, such as ResNet50 and ViT-B/16, exhibit higher resilience to the smoothing introduced by differential privacy mechanisms. This resilience can be attributed to the models' enhanced capacity to learn robust feature representations, which can better withstand the perturbations caused by smoothing. This finding aligns with existing literature that suggests a direct correlation between model complexity and its ability to generalize from noisy data Kaplan et al. (2020). The significant performance gap between our method and the traditional DP-SGD approach, even at higher $\epsilon$ values, underscores the effectiveness of our approach in balancing the privacy-utility trade-off. Traditional DP-SGD methods often suffer from a substantial reduction in utility due to the added smoothing, which is necessary to ensure privacy. Our method, however, achieves competitive accuracy with a much lower privacy budget, demonstrating a more efficient utilization of the privacy budget.

**Impact of $\epsilon$ on Performance**   The parameter $\epsilon$ in differential privacy quantifies the trade-off between privacy and utility. Our results show that while increasing $\epsilon$ improves the performance of the DP-SGD method, it still lags behind our method with a lower $\epsilon$. This suggests that our method can

achieve higher accuracy even under stricter privacy constraints, making it more suitable for applications where both high utility and stringent privacy guarantees are required. Our method can allocate the budget at instance level, which will not waste privacy on the samples in dense PDF regions and degrade the performance. The smoothing effect for the sparse samples is economically allocated as the backbone can provide more precise representation, resulting in less performance degradation as the privacy budget increases.

### 4.4 RESULT AND DISCUSSION ON CHEXPERT

In the realm of medical image analysis, ensuring patient privacy while maintaining high model performance is a paramount concern. The CheXperf dataset, a comprehensive collection of thoracic condition images, serves as an excellent benchmark for evaluating privacy-preserving techniques. Our proposed instance-level PDF smoothing model exhibits significant advancements over existing methods such as DP-SGD and non-private models.

The experimental results depicted in Figure 2 reveal several critical insights. Our proposed PDF smoothing model consistently outperforms DP-SGD under the same backbone and experiment settings and has a narrow gap with non-private model. This indicates that our method effectively balances the trade-off between privacy and utility. As expected, higher values of $\epsilon$ correlate with improved model accuracy due to reduced smoothing factor. This is a well-documented phenomenon in differential privacy literature. However, our adaptive DP model demonstrates superior performance even at lower $\epsilon$ values, underscoring its robustness. For thoracic conditions with privacy budgets set to $\epsilon = 2$ and $\epsilon = 8$, our model achieves an average testing AUC of 85% across all five labels. In contrast, DP-SGD lags behind, achieving approximately 70%. This significant margin underscores the efficacy of our adaptive approach in maintaining high accuracy while preserving privacy.

**Instance-Level Privacy Allocation**  A key strength of our model lies in its instance-level privacy allocation. By dynamically adjusting the privacy budget for each data instance based on its probability density, we ensure that samples in sparse regions receive higher privacy protection. This targeted approach is more economical and effective compared to uniform noise addition strategies. Medical images often contain critical and unique information that might appear infrequently (i.e., in sparse regions of the data distribution). Our model enhances privacy for these samples without significantly compromising their utility. This is particularly vital in medical contexts where rare conditions need accurate representation and analysis. Conversely, in dense regions where data points are abundant, excessive smoothing injection can severely disrupt the model's learning process. Our method strategically injects less smoothing in these areas, preserving the inherent structure and relationships within the data. This selective smoothing ensures that the overall performance of the model remains high.

**Class Disparity Improved by Instance level Probability Smoothing**  In Figure 3, we compare class disparities between our private models (at $\epsilon$=8) and non-private baselines. Both model types show similar disparities across subgroups, with private models not systematically worse in terms of AUC disparities. Notably, smaller subgroups, such as *Atelectasis*, suffer more performance degradation due to their sparse feature distribution. Contrary to prior works, our private models demonstrate significant improvement over the DP-SGD method, achieving better group fairness outcomes than non-private baselines. This suggests that increased disparities in private models observed previously are not inherent. Our instance-level PDF smoothing model effectively balances privacy and performance, protecting sparse samples economically while minimizing disruption in dense regions. Future work could extend this method to other medical datasets and real-time clinical systems, and explore adaptive mechanisms for dynamic privacy budget adjustments.

## 5 CONCLUSION

In conclusion, we introduce a novel instance-level PDF smoothing model that enhances privacy preservation while maintaining high performance in medical image analysis. Our approach offers profound theoretical insights into sample complexity, smoothing factors, and error bounds for achieving privacy budgets. Our experimental results demonstrate the efficacy of our model in balancing privacy and utility, dynamically adjusting privacy parameters to protect samples in sparse regions

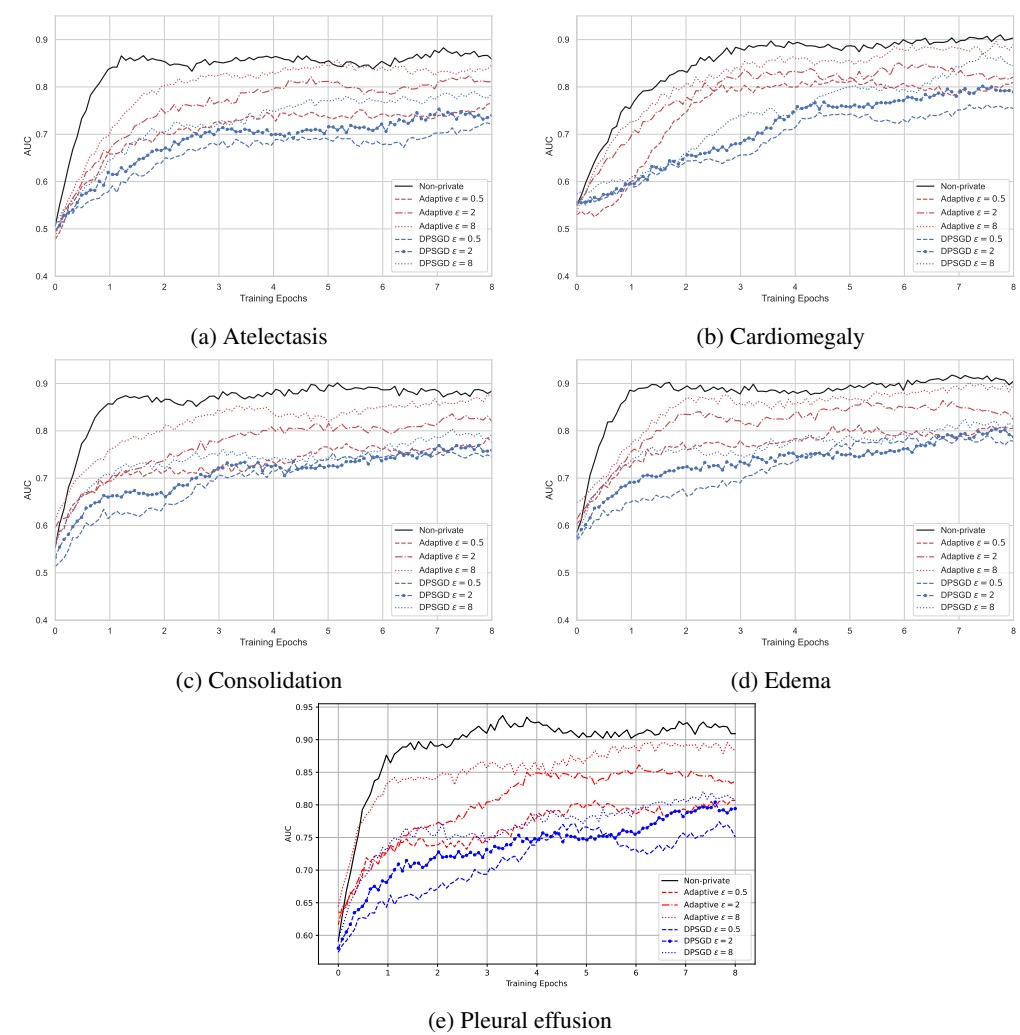

(a) Atelectasis

(b) Cardiomegaly

(c) Consolidation

(d) Edema

(e) Pleural effusion

Figure 3: Comparison between non-private model, DP-SGD and our adaptive DP model under different privacy budget settings. The x-axis the training iterations in epochs and the y-axis is the model performance in AUC.

Table 1: Performance of Different Algorithms on CIFAR-10

| DP Algorithm | ResNet18 | ResNet50 | ViT-B/16 |
|---|---|---|---|
| DPSGD ($\epsilon = 3$) | 77.8 | 81.5 | 92.4 |
| DPSGD ($\epsilon = 8$) | 82.9 | 84.1 | 94.0 |
| Ours ($\epsilon = 3$) | 85.2 | 89.9 | 94.2 |
| Ours ($\epsilon = 8$) | 90.2 | 92.9 | 95.2 |
| Non-Private | 93.2 | 94.9 | 97.1 |

and addressing class disparities in medical datasets. The adaptive DP model not only preserves the utility of critical medical information but also improves overall model fairness. Our findings open several avenues for future research, including extending to other medical datasets and integrating with real-time clinical systems. Our instance-level PDF smoothing model represents a significant step forward in privacy-preserving medical image analysis, offering a robust, comprehensive solution to the challenges posed by differential privacy. We hope our work inspires further innovation, contributing to safer and more effective healthcare technologies. Limitations list in Appendix A.4.

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

## A APPENDIX

You may include other additional sections here.

### A.1 PROOF OF THEOREM 1

*Proof.* Zero-Concentrated Differential Privacy (zCDP) is defined such that a mechanism $M$ satisfies $\rho$-zCDP if the privacy loss random variable $L$ satisfies $\mathbb{E}[e^L] \le e^\rho$. The privacy loss $L$ is given by $L = \log\left(\frac{M(D_1)}{M(D_2)}\right)$, where $D_1$ and $D_2$ are neighboring datasets differing by one element.

Consider the Gaussian mechanism $M$ that adds smoothing $N_i \sim \mathcal{N}(0, \sigma^2)$ to each sample $x_i$.

The privacy loss $L$ for the Gaussian mechanism is given by $L = \log\left(\frac{p(M(D_1))}{p(M(D_2))}\right)$. For two neighboring datasets $D_1$ and $D_2$, the difference in the KDE estimates is small, and the privacy loss $L$ can be approximated using the Rényi divergence.

The Rényi divergence $D_\alpha$ between two distributions $P$ and $Q$ is defined as:

$$D_\alpha(P\|Q) = \frac{1}{\alpha - 1} \log\left(\int p(x)^\alpha q(x)^{1-\alpha}\, dx\right)$$

For $\alpha > 1$, the Rényi divergence is a measure of the distance between two distributions.

The $\rho$-zCDP condition can be expressed in terms of Rényi divergence:

$$D_\alpha(P\|Q) \le \rho$$

For $\alpha = 2$, the Rényi divergence simplifies to:

$$D_2(P\|Q) = \log\left(\int p(x)^2 q(x)^{-1}\, dx\right)$$

For the Gaussian mechanism, the smoothing $N_i \sim \mathcal{N}(0, \sigma^2)$ ensures that the privacy loss $L$ is controlled. The Rényi divergence for the Gaussian mechanism is given by:

$$D_2(P\|Q) \le \frac{\Delta^2}{2\sigma^2}$$

where $\Delta$ is the sensitivity of the function being perturbed.

The sensitivity $\Delta$ is the maximum change in the PDF estimate due to adding or removing one sample.

To achieve $\rho$-zCDP, we need $D_2(P\|Q) \le \rho$. Substituting the Rényi divergence for the Gaussian mechanism, we get:

$$\frac{\Delta^2}{2\sigma^2} \le \rho$$

Solving for $\sigma^2$, we have:

$$\sigma^2 \ge \frac{\Delta^2}{2\rho}$$

To account for the confidence bound $\delta$, we use the fact that the tail probability of the Gaussian distribution is controlled by $\delta$:

$$\Pr[L > \rho] \le \delta$$

Using the Chernoff bound, we obtain:

$$\sigma^2 \ge \frac{2\ln(1/\delta)}{\rho}$$

$\square$

## A.2 Proof of Theorem 2

*Proof.* The change in Rényi divergence $\Delta D_\alpha$ can be bounded using the properties of the PDF estimate:

$$\Delta D_\alpha = \frac{1}{\alpha - 1} \left( \log \left( \int \hat{p}_{-k}(x)^\alpha Q(x)^{1-\alpha} \, dx \right) \right.$$
$$\left. - \log \left( \int \hat{p}(x)^\alpha Q(x)^{1-\alpha} \, dx \right) \right)$$

For large $n$, the change in the KDE estimate due to removing a single sample can be approximated as:

$$\hat{p}_{-k}(x) \approx \hat{p}(x) - \frac{1}{n} K_h(x - x_k)$$

The change in Rényi divergence can be bounded using the stability of the KDE estimate:

$$\Delta D_\alpha \leq \frac{1}{\alpha - 1} \log \left( 1 + \frac{C}{n} \right)$$

where $C$ is a constant that depends on the kernel and the bandwidth $h$.

To ensure $\rho$-zCDP, we need:

$$\frac{1}{\alpha - 1} \log \left( 1 + \frac{C}{n} \right) \leq \rho \alpha$$

Simplifying this inequality:

$$\log \left( 1 + \frac{C}{n} \right) \leq \rho(\alpha - 1)$$

For small $C/n$, we can use the approximation:

$$\frac{C}{n} \leq \rho(\alpha - 1)$$

Solving for $n$:

$$n \geq \frac{C}{\rho(\alpha - 1)}$$

To derive the constant $C$, we need to consider the maximum possible change in the KDE estimate due to removing a single sample.

Maximum Change in KDE Estimate:

$$\Delta \hat{p}(x) \approx \frac{1}{n} K_h(x - x_k)$$

Bounding the Maximum Change, the maximum value of the kernel function $K_h(x)$ is:

$$\frac{1}{\sqrt{2\pi h^2}}$$

Therefore, the maximum change in the KDE estimate is:

$$\Delta \hat{p}(x) \leq \frac{1}{n} \cdot \frac{1}{\sqrt{2\pi h^2}}$$

Formulating $C$: The constant $C$ can be formulated as:

$$C = \frac{1}{\sqrt{2\pi h^2}}$$

$\square$

### A.3 PROOF OF THEOREM 3

*Proof.* To achieve $\rho$-zCDP, smoothing $N_i \sim \mathcal{N}(0, s_i^2 \cdot \frac{2\ln(1/\delta)}{\rho})$ is added to each sample $x_i$:

$$x_i' = x_i + N_i$$

The privacy preserved PDF $\hat{p}'(x)$ is:

$$\hat{p}'(x) = \frac{1}{n} \sum_{i=1}^{n} K_h(x - x_i')$$

The classification error is affected by the smoothing added to the PDF. It can be bounded by the PDF margin between different classes:

$$\text{Err}(f) \le \Pr\left(|\hat{p}(x) - \hat{p}'(x)| > m\right)$$

The smoothing variance $\sigma^2$ is:

$$\sigma^2 = \frac{2\ln(1/\delta)}{\rho}$$

Thus, the classification error bound is given by:

$$\text{Err}(f) \le \Pr\left(|\hat{p}(x) - \hat{p}'(x)| > m\right) \le \Pr\left(|N_i| > m\right)$$

Using the properties of the Gaussian distribution:

$$\Pr\left(|N_i| > m\right) \le 2\exp\left(-\frac{m^2}{2\sigma^2}\right)$$

Substituting $\sigma^2 = \frac{2\ln(1/\delta)}{\rho}$, we get:

$$\Pr\left(|N_i| > m\right) \le 2\exp\left(-\frac{m^2 \rho}{4\ln(1/\delta)}\right)$$

Therefore, the classification error bound is:

$$\text{Err}(f) \le 2\exp\left(-\frac{m^2 \rho}{4\ln(1/\delta)}\right)$$

$\square$

### A.4 LIMITATIONS

While our proposed framework demonstrates significant advancements in balancing privacy and utility in differentially private (DP) deep learning models, several limitations warrant discussion:

1. **Generalization Across Domains:** Although our empirical studies on the CheXpert dataset show promising results, the generalization of our method to other domains and data types remains to be thoroughly validated. Medical imaging datasets, like CheXpert, may possess unique characteristics that are not representative of other datasets, potentially limiting the applicability of our findings to different domains such as natural language processing or other forms of visual data.

2. **Scalability and Computational Overhead:** Our instance-level smoothing mechanism, while effective, introduces additional computational overhead that may not be scalable for extremely large datasets or models with billions of parameters. The increased computational requirements could hinder the practical deployment of our framework in resource-constrained environments.

3. **Hyperparameter Sensitivity:** The success of our method heavily relies on the careful tuning of hyperparameters related to instance-level smoothing and noise injection. Identifying the optimal settings for these parameters can be challenging and may require extensive experimentation, which could limit the ease of adoption for practitioners with limited resources or expertise in hyperparameter optimization.

4. **Impact on Training Time:** The necessity to balance privacy and utility often comes with a trade-off in training time. Our method, which includes additional smoothing processes and noise injection, may result in longer training times compared to non-DP models. This could be a critical limitation for applications requiring rapid model development and deployment.

5. **Potential for Subpopulation Fairness:** Despite our efforts to mitigate accuracy disparities across subpopulations, the inherent complexity of ensuring fairness in DP models means that some level of disparity may still persist. Future work is needed to develop more robust methods for guaranteeing fairness across diverse and heterogeneous subpopulations.

6. **Theoretical Assumptions:** Our theoretical insights and error bounds are based on certain assumptions regarding sample complexity and instance-level smoothing factors. The practical validity of these assumptions in real-world scenarios needs further empirical validation across a broader range of datasets and model architectures.

7. **Privacy Budget Allocation:** The allocation of the privacy budget in DP models is a critical factor that influences the overall performance. Our framework provides guidelines for optimizing this balance, but the practical implementation of these guidelines can be complex and may not always achieve the desired outcomes in every context.

Addressing these limitations in future research will be crucial for advancing the practical applicability and robustness of differentially private deep learning models, ensuring that they can meet the dual challenges of high accuracy and fairness across a wide range of applications.

