# OpenReview forum: "Instance-Level Smoothing for Enhanced Privacy in Deep Learning: Theoretical Insights and Empirical Validation"
_ICLR.cc/2025/Conference — ICLR 2025 Conference Withdrawn Submission_

### Official Review · Reviewer_6RE2 · 2024-10-31

**Soundness:** 1
**Presentation:** 1
**Contribution:** 3
**Rating:** 3
**Confidence:** 3

**Summary:**

This paper proposes an alternative to DP-SGD for differentially private fine-tuning of public pre-trained models, which is based on instance-level smoothing and results in an improved utility-privacy trade-off.

**Strengths:**

This work combines existing ideas in an interesting way, addressing an important issue in private deep learning.

**Weaknesses:**

The paper is lacking clarity in the following ways:
1) It remains unclear that the method requires a public feature extraction model until Section 3.3. I would expect this important information to be already mentioned in the abstract and introduction, as well as in the conclusion (as a limitation of your approach in comparison to DP-SGD).
2) The difference between sample and instance is unclear. Do you use them as synonyms? Please clarify.
3) You mention again and again that your method achieves a good privacy-utility trade-off, but in the beginning of the paper it already should be clear how you achieve this. For example, clarify early on that you use instance-level privacy budget allocation.
4) You promise "more efficient training" (line 184). The term is ambiguous. I expected faster training speeds, however, it seems that you were merely referring to higher accuracy.
5) Some abbreviations and variables are not explained. For example, while KDE is a popular abbreviation, it still makes sense to introduce it explicitly. Moreover, the delta in Theorems 1 and 3 is not explained. It is only mentioned later that it is a different delta than in Sections 2 and 4.
6) The figures need improvement. The labels and numbers in Figure 1 are too small. Figure 3e has a differently scaled y-axis than the other subplots. Figure 2 is incorrectly referenced in Section 4.4. (I suppose the reference should point to Figure 3?)
7) Table 1 is not referenced anywhere.
8) You claim that your method reduces class disparities, however, Figure 3 is not ideal to show this. I would suggest an explicit comparison instead.
9) After Theorem 3, you claim that a higher rho leads to more smoothing and thus lower classification accuracy. But a higher rho means lower privacy, i.e., less smoothing.

Overall, while the idea seems interesting, the soundness and presentation of the paper needs to be significantly improved. Also make sure that the paper does not include any remnants of the template (e.g., "You may include other additional sections here." in line 650).

**Questions:**

Can your method be also applied for regression tasks? What if no public pre-trained model is available for the task in question? (I would like to see these discussions in the limitation section)

---

### Official Review · Reviewer_3XtR · 2024-11-01

**Soundness:** 1
**Presentation:** 3
**Contribution:** 2
**Rating:** 5
**Confidence:** 3

**Summary:**

This work investigates DP training for adaptive kernel metric representation learning. Unlike DP-SGD, the proposed approach adds noise to individual data point embeddings instead of aggregated gradients. Specifically, the method maps each data point to an embedding using a pre-trained backbone model and subsequently perturbs each embedding with Gaussian noise. The sensitivity for each sample varies and is estimated using the equation in line 266. As a result, this sample-specific sensitivity is both an approximation and data-dependent.

**Strengths:**

1. The discussion on the impact of inter-class discrepancies on accuracy is compelling.
2. The authors implement the proposed algorithm on standard benchmarks as well as a real-world medical dataset.
3. The paper is well-structured and clearly written.

**Weaknesses:**

1. The local sensitivity depends on individual samples and is not released privately, this invalidates the DP guarantee.

To strengthen the privacy claims, the authors could conduct empirical privacy attacks and compare the empirical protection offered by this method against DP-SGD.

2. As mentioned in lines 263-268, the local sensitivity $s_{i}$ for the $i_{th}$ sample is an approximation. Can the authors provide bounds on the approximation error or conduct experiments to showcase the magnitude of this error?

Minor Suggestion:
To improve readability, consider assigning indexes to important equations for easier reference.

**Questions:**

Please refer to 'Weaknesses'.

---

### Official Review · Reviewer_hPoS · 2024-11-03

**Soundness:** 1
**Presentation:** 2
**Contribution:** 1
**Rating:** 3
**Confidence:** 5

**Summary:**

The paper proposes an instance-level kernel smoothing method for training deep learning models with differential privacy, via estimating the pdf of the training dataset.

**Strengths:**

1. Deep learning with differential privacy is an important topic to this field.

**Weaknesses:**

1. There is a discrepancy in the reported results. In the proposed solution, each input record is perturbed with a Gaussian noise (referred to as smoothing). This is analogous to the local DP model. As established in prior work, notably by Kairouz et al. in “Discrete distribution estimation under local privacy” (ICML 2016), local DP usually leads to a substantial reduction in accuracy compared to centralized DP due to the higher noise levels required. However, the results in Table 1 of the paper show that the proposed method outperforms a centralized-DP baseline, which is counterintuitive. Clarifying why the proposed solution yields such unexpectedly high performance would strengthen the paper significantly.

2. It is not clear why the proposed method satisfies DP. Effectively, the scale of the noise used by the proposed method is dependent on one record of the input dataset. This leads to the notion of local sensitivity, rather than global sensitivity. It is well-known that injecting noises according to local sensitivity could violate differential privacy, since the noise scale itself is considered private (see Smooth Sensitivity and Sampling in Private Data Analysis by Nissim et al. for more details).

**Questions:**

Please clarify Weakness 1 & 2.

---

### Official Review · Reviewer_NP36 · 2024-11-04

**Soundness:** 1
**Presentation:** 2
**Contribution:** 2
**Rating:** 3
**Confidence:** 4

**Summary:**

The paper proposes to learn private projections from a pre-trained feature space to KDE space, followed by k-NN in the KDE space for classification. The paper also proposes a special kernel appropriate for high-dimensional representations. Finally, the paper argues that the method attains better performance than standard DP-SGD of pre-trained vision models on CIFAR-10 and CheXpert datasets at the same level of privacy.

**Strengths:**

The paper proposes a method for efficient private fine-tuning using public pre-trained feature extraction models. The intuition that "samples with sparse distributions in their feature space, based on pre-trained large models, are more prone to privacy leakage" makes sense, at it is a good idea to exploit it.

**Weaknesses:**

The paper has a critical issue:

*Incorrect privacy analysis.* The analysis does not show that the method satisfies DP or zCDP. If analyzed as standard Gaussian mechanism, the sensitivity must be global, i.e., use $C$ from line 753 instead of $s_i$. The quantity $s_i$ is not local sensitivity. Local sensitivity is defined as $\sup_{S \simeq S'} ||f(S) - f(S')||$, where $S$ is fixed to the current dataset, and $S'$ is any other neighboring dataset ([Vadhan, 2016](https://privacytools.seas.harvard.edu/files/complexityprivacy_1.pdf), Section 3). The approximate computation in line 265 only considers $S'$ which differ from $S$ by exclusion of a given example $i$, which is not sufficient for computing local sensitivity under either of the standard add/remove or substitution relations. Even if considering remove-only relation (which is technically possible, but not sufficient to obtain standard operational guarantees of DP), local sensitivity would have to be maximum over records $\max_{j \in [n]} |\hat p(x) - \hat p_{-j}(x)|$. Besides, using local sensitivity requires additional algorithmic consideration (e.g., smooth sensitivity or propose-test-release frameworks, see (Vadhan, 2016) and references therein). Therefore, I do not think the method satisfies DP or zCDP.

The other issues are not as critical, but significant still:

*Unclear presentation.* The method is not sufficiently and clearly detailed. Notions are often used in text before being defined, and terminology is likely inconsistent because of that. For instance:
- What exactly is a projection network, i.e., what is the entire function being applied to samples?
- What is bold K in Eq. 1? Where is bold W used in Eq. 1? How are batches handles inside bold K?

*Novelty.* Beyond issues with correctness of the privacy analysis and presentation, similar ideas were proposed by [Tramer & Boneh, 2021](https://arxiv.org/abs/2011.11660), and the method (after corrections), should compare to such prior approaches.

**Questions:**

- How exactly were the pre-trained vision models adapted for DP-SGD? E.g., batch norm was changed to group norm?

---

### Note · Authors · 2024-11-13

I have read and agree with the venue's withdrawal policy on behalf of myself and my co-authors.